# Foliar spraying exogenous ABA resists chilling stress on adzuki beans (*Vigna angularis*)

**Hongtao Xiang**[1,2☯], **Xiaoyan Liang**[3☯], **Shiya Wang**[3☯], **Xueyang Wang**[2], **Ning He**[2], **Xiaohui Dong**[2], **Deming Wang**[1,2], **Shuqiang Chen**[2], **Qiulai Song**[2], **Yuqiang Liu**[2], **Qingjuan Wang**[4], **Wan Li** [iD][2]*

**1** Suihua Branch, Heilongjiang Academy of Agricultural Machinery Sciences, Suihua, China, **2** Heilongjiang Academy of Agricultural Sciences, Harbin, China, **3** College of Agriculture, Heilongjiang Bayi Agriculture University, Daqing, China, **4** Qiqihar Agricultural Technology Promotion Center, Qiqihar, China

☯ These authors contributed equally to this work.
* zpszls@aliyun.com

**Data Availability Statement:** All relevant data are within the manuscript and its Supporting Information files.

## Abstract

Adzuki bean, an important legume crop, exhibits poor tolerance to low temperatures. To investigate the effect of exogenous abscisic acid (ABA) on the physiological metabolism and yield resistance of adzuki bean under low-temperature stress, we conducted a potted experiment using Longxiaodou 4 (LXD 4) and Tianjinhong (TJH) as test materials and pre-sprayed with exogenous ABA at flowering stage continuously for 5 days with an average of 12˚C and an average of 15˚C, respectively. We found that, compared with spraying water, foliar spraying exogenous ABA increased the activities of antioxidants and the content of non-enzymatic antioxidants, effectively inhibited the increase of malondialdehyde (MDA), hydrogen peroxide ($H_2O_2$) content, $O_2^{\cdot-}$ production rate. Exogenous ABA induced the activation of endogenous protective mechanisms by increasing antioxidant enzymes activities such as superoxide dismutase (SOD), peroxidase (POD), and catalase (CAT), as well as elevated levels of non-enzymatic antioxidants including ascorbic acid (ASA) and glutathione (GSH). Moreover, the yield loss of 5.81%-39.84% caused by chilling stress was alleviated by spraying ABA. In conclusion, foliar spraying exogenous ABA can reduce the negative effects of low-temperature stress on the yield of Adzuki beans, which is essential to ensure stable production of Adzuki beans under low-temperature conditions.

## Introduction

Adzuki bean, one of the most important legume crops for human consumption, is cultivated in China, Korea, Japan, the Philippines, and other Southeast Asian countries [1]. It possesses biological characteristics of light, temperature, and infertile but does not tolerate low temperatures and is sensitive to chilling [2]. China is the world's largest producer and primary exporter of adzuki beans. According to statistics from 2016 to 2018, the sown area of the country was above 180 thousand hectares; among them, the sown area of Heilongjiang Province in the northeast accounted for about 50% of the country (www.stats.gov.cn), which is the largest production base in China.

**Funding:** This research was supported in part by the Heilongjiang Key R&D Program Project (GA21B009-14), and China Agriculture Research System (CARS-08-G8) Project for the financial support. The funder wrote the manuscript.

**Competing interests:** The authors declare that they have no known competing financial interests or personal relationships that could have appeared to influence the work reported in this paper.

The optimum growth temperature range for adzuki beans lies between 20˚C and 24˚C. Temperatures below this threshold induce low-temperature stress or even chilling damage [3]. Low temperature adversely affects plant growth and metabolism, causing a large accumulation of reactive oxygen species (ROS) free radicals within plants [4]. While plants possess defense mechanisms capable of eliminating some ROS molecules promptly, those that remain unremoved accumulate over time within plants' cellular structures, leading to lipid peroxidation damage on cell membranes. This disruption impairs normal physiological functions and metabolic pathways in crops, ultimately reducing crop yields. In recent years, meteorological disasters such as extreme weather have occurred frequently, and sudden cold damage has become an essential abiotic adversity stress that restricts agricultural production in Heilongjiang province [2]. Among various factors affecting crop productivity, chilling damage is one of the most important [5], which interferes with crops' physiological and biochemical processes and restricts crop yield and quality [6]. From sowing to maturity, legume crops are frequently subjected to periodic low-temperature cold stress, mainly at flowering and podding, which tends to reduce yield and quality [7]. Because flowering serves as a critical period for the coexistence of nutritive and reproductive growth, the occurrence of cold stress at this time reduces the quantity and quality of pollen in the stigma, leading to pod abscission and lower yields [8, 9].

Applying plant growth regulators is one of the crucial measures to alleviate abiotic stress in crops [10]. Abscisic acid (ABA), a sesquiterpene-structured plant hormone, plays a significant role in controlling plant growth, inhibiting seed germination, promoting senescence, and responding to low-temperature adversity stress [11]. Exogenous ABA has an induced effect to improve the cold resistance of crops and can increase the oxidase activity of crops while slowing down the accumulation of membrane lipid peroxide MDA [12], promoting the accumulation of free proline [13] and changing the content of endogenous hormones in crops, making the endogenous ABA content increase, while decreasing the GA content, thus improving the crop's cold resistance [10, 14]. However, studies on the effects of low-temperature stress and sprayed exogenous ABA during the flowering stage on adzuki bean leaf physiological indicators and yield are lacking.

To fill this knowledge gap, we conducted an artificially simulated low-temperature experiment aiming to investigate the physiological mechanism underlying the impact of low temperature on adzuki beans during flowering, as well as evaluate the alleviating effect of exogenous ABA on cold damage. This study is expected to provide a theoretical basis for cold-resistant adzuki bean production in the northern hemisphere.

## Materials and methods

### Experimental design

The popular cold-tolerant and cold-sensitive adzuki bean varieties, LXD 4 and TJH were used as experimental materials. Exogenous ABA was provided by China National Practical Bean Technology System, purchased from Sigma Company. The experiment was carried out at the potted planting field of the Institute of Crop Cultivation and Tillage, Heilongjiang Academy of Agricultural Sciences. Adzuki bean seeds were planted in each resin pot (diameter of 30 cm and height of 25 cm) filled with about 16 kg of soil. This experiment has six treatments, as shown in Table 1, and the temperature change during the day is shown in Fig 1. When plants grow to flowering stage (day 51 after sowing), conduct foliar spraying of exogenous ABA (concentration of 20 mg×L$^{-1}$ and a spraying rate of 225 L×hm$^{-2}$), and the pots were entered into the artificial climate room with different low temperatures for chilling treatment. The duration is 1, 2, 3, 4, and 5 days. Each treatment was sampled separately and immediately placed in liquid nitrogen, then stored in a -80˚C refrigerator to determine physiological indicators.

**Table 1. Test design scheme.**

| Treatment code | Leaf spray treatment | Temperature treatment |
|---|---|---|
| T1 | Spray ABA | Average 12°C |
| T2 | Spray water | Average 12°C |
| T3 | Spray ABA | Average 15°C |
| T4 | Spray water | Average 15°C |
| T5 | Spray ABA | Normal temperature |
| CK | Spray water | Normal temperature |

## Determination of MDA, $H_2O_2$ content and $O_2^{\cdot-}$ production rate

The content of MDA (malondialdehyde) was determined using the thiobarbituric acid (TBA) method [15]. 0.5 g of the samples were ground into homogenate in 10 ml phosphate buffer (0.05 mM PBS, pH 7.8), centrifuge at 6000 r/min for 20 min., Add 2 ml of 0.6% TBA to 1 ml of the supernatant, mix well, and boil in a 100°C boiling water bath for 15 min, cool quickly with cold water. Centrifuge at 4000 r/min for 20 min; take supernatant to determine OD value under 450 nm, 532 nm, and 600 nm.

The hydroxylamine oxidation method determined the $O_2^{\cdot-}$ production rate [16]. The 0.5 mL extract was obtained by adding 0.5 ml PBS (0.05M, pH 7.8), 1 mL 10 mM hydroxylamine hydrochloride solution, shake well, and leave to stand for about 1h at 25°C. For chlorophyll extraction, add 2 mL ether to the solution followed by 1 mL 7mM p-aminobenzene sulfonic acid and then 1mL 7 mM α-naphthylamine. Mix well, swirl, and leave to stand for 20 min at 25°C. Centrifug the mixture at 3000 r/min for 3 min and determine the OD value at 530 nm.

The content of $H_2O_2$ (hydrogen peroxide) was determined by the potassium iodide method [17]. 1 g of the sample was added to 0.5 mL of 0.1% TCA. The homogenate obtained was centrifuged at 19000 r/min for 20 min. 0.5 ml of the supernatant was added to 2 mL of 1 m KI solution and 0.5 mL of 100 mM potassium sulfate buffer. The dark reaction was performed for 1 hr, after which 0.1% TCA was used as a reference to determine the OD value at 390 nm.

## Determination of the activities of superoxide dismutase (SOD), peroxidase (POD), and catalase (CAT)

0.5 g of frozen leaf samples were ground into homogenate in 10 mL of 50 mM phosphate buffer (pH 7.8) on an ice bath and centrifuged at 4°C at 12000 ×g for 20 min. The supernatant

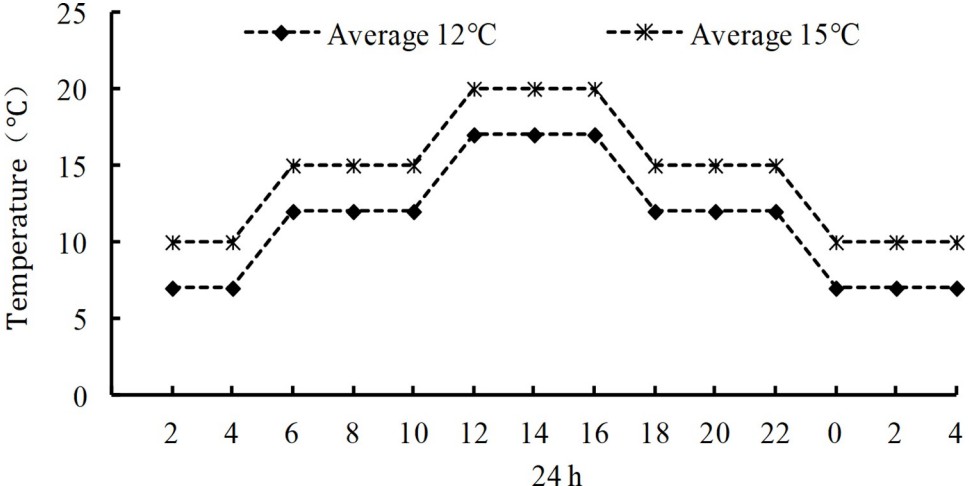

**Fig 1. Changes in temperature during one day.**

was used to determine enzyme activity. SOD (superoxide dismutase) activity was determined by the nitrogen blue tetrazolium (NBT) method at 560 nm [18]. POD (peroxidase) was determined by the oxidation rate of guaiacol at 470 nm [19]. CAT (catalase) was determined by a decrease in absorbance per minute at 240 nm, outlined by Fu and Huang [19].

### Determination of enzyme activity in AsA-GSH cycle

About 0.5 g of frozen leaf samples were ground into homogenate in 10mL of 50 mM phosphate buffer (pH 7.8) on an ice bath and centrifuged at 4˚C at 12000 ×g for 20 min. The supernatant was used to determine enzyme activity. APX (ascorbate peroxidase) was measured at 290 nm using the hydrogen peroxide method [20]. GR (glutathione reductase) was determined at 340 nm according to the method described by Zhu [21] and Wheeler [22]. DHAR (dehydroascorbate reductase) was determined at 290 nm according to the method of Murshed [23]. MDHAR (monodehydroascorbate reductase) was determined using the kit method (Solarbio company).

### Determination of AsA-GSH cycle products and substrate content

About 1 g of leaf sample was ground in 5 mL of 5% metaphosphoric acid. The homogenate was then centrifuged at 8000 g for 15 min. The supernatant was collected to determine AsA-GSH cycle products and substrate content. AsA (ascorbic acid) and DHA (dehydroascorbate) were determined using the method described by Zhang and Kirkham [24]. The contents of reduced GSH (glutathione) and GSSG (glutathiol) were determined using the dithionitrobenzoic acid method [25].

### Statistical analysis

Microsoft Excel 2013 and SPSS 22.0 were used to analyze the collected data's one-way ANOVA. Duncan test ($p < 0.05$) was used to evaluate the difference within treatments, and the significant differences among different materials were determined. Moreover, the Structural Equation Modeling (SEM) framework with AMOS 22.0 software (Small Waters Corporation, Chicago, IL, USA) was applied to estimate further the direct and indirect effects of exogenous ABA, low temperature, and chilling time on the yield of Adzuki beans. All figures were prepared using Origin software 2018.

## Results

### Effect of exogenous ABA on MDA content in adzuki beans leaves under chilling stress

As shown in Fig 2, the MDA content in the adzuki bean leaves showed an increasing trend after the low-temperature treatment during the flowering stage. The content of MDA at an average 12˚C (T2) treatment was higher than the average 15˚C (T4). The MDA content followed an increase with the extension of the low-temperature treatment time. When treated for 5 days, compared with CK, the T2 and T4 treatments of LXD 4 significantly increased MDA by about 4 and 3.8 times, respectively, with corresponding MDA values of 42.43 nmol·g$^{-1}$ and 40.03 nmol·g$^{-1}$. The T2 and T4 treatments of TJH significantly increased 47.36 nmol·g$^{-1}$ and 43.91 nmol·g$^{-1}$, respectively. Exogenous ABA can significantly inhibit the increase of MDA content under low-temperature conditions. The analysis of variance showed that after treatment for 1 to 5 days, both LXD 4 and TJH showed that T1 was significantly lower than T2; T3 was significantly lower than T4.

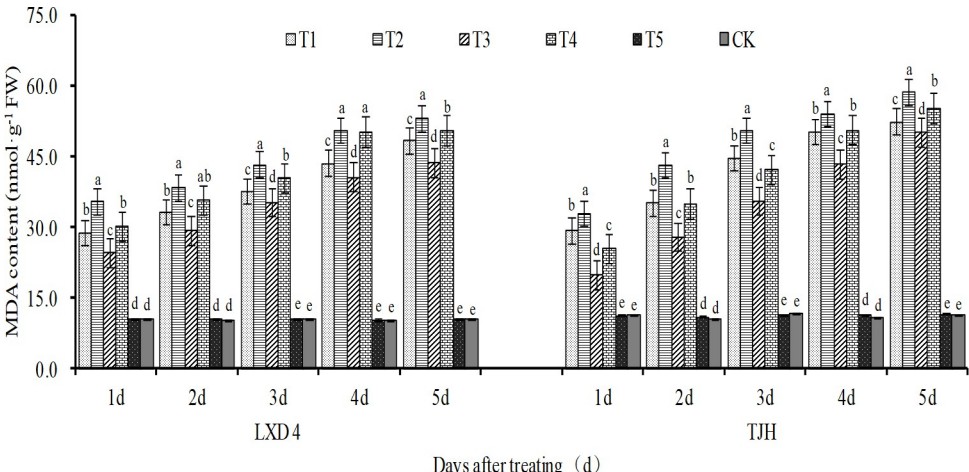

**Fig 2. Effect of exogenous ABA on MDA content of adzuki beans leaves under chilling stress at flowering stage;
CK, plant grown in natural environment + spraying H2O; T1, plant grown in average 12˚C+ spraying exogenous
ABA; T2, plant grown in average 12˚C+ spraying H2O; T3, plant grown in average 15˚C+ spraying exogenous
ABA; T4, plant grown in average 15˚C+ spraying H2O; T5, plant grown in natural environment + spraying ABA.**
Note: Vertical bars represent standard error of the mean (n = 3). Different letters above the bars within each panel
indicate significant differences across treatments according to the Duncan test (0.05).

## Effect of exogenous ABA on $H_2O_2$ content and $O_2^{·−}$ production rate in adzuki beans leaves under chilling stress

The low temperature during the flowering stage leads to the increase of $H_2O_2$ content in the
leaves of the adzuki bean. Low-temperature treatment for 1 to 5 days, compared with CK, the
$H_2O_2$ content of T2 and T4 treatments increased significantly of LXD 4, and the changing cir-
cumstances of TJH were the same. Under low-temperature conditions, foliar spraying with
exogenous ABA can effectively inhibit the production of $H_2O_2$ and reduce its damage. For LXD
4, when treated for 3 to 5 days, T1 was significantly lower than T2 by 6.95%, 9.07%, and 9.03%,
T3 was significantly lower than T4 by 8.16%, 9.18%, and 11.42%. For TJH, treatment for 1 to 5
days, T1 was significantly lower than T2 by 7.00%, 5.41%, 5.11%, 14.21%, and 10.03%, T3 was
significantly lower than T4 by 3.68%, 2.86%, 8.18%, 16.19%, and 6.36% (Fig 3A).

As shown in Fig 3B, the $O_2^{·−}$ production rate of adzuki bean leaves showed a linear increase
under different low-temperature treatments during the flowering stage. In low-temperature
treatment for 1 to 5 days, of bothT2 and T4 treatments of LXD 4 significantly increased the
$O_2^{·−}$ production rate compared with CK, and the phenomenon of TJH was the same. In addi-
tion, in each treatment day, T2 treatment was higher than T4 treatment, whether it was LXD 4
or TJH, and this indicates that the lower the temperature, the faster the $O_2^{·−}$ production rate
and the greater the damage to adzuki beans. Spraying exogenous ABA can inhibit the produc-
tion rate of $O_2^{·−}$ in leaves; the analysis of variance showed that after treatment for 1 to 5 days,
both LXD 4 and TJH showed that T1 was significantly lower than T2; T3 was significantly
lower than T4.

## Effect of exogenous ABA on SOD, POD, and CAT activities in adzuki beans leaves under chilling stress

The activities of SOD, POD, and CAT in the leaves showed an increasing trend after low-tem-
perature treatment of adzuki beans at the flowering stage. Spraying exogenous ABA can fur-
ther increase the activities of these three antioxidant enzymes (Fig 4). In addition, the cold-

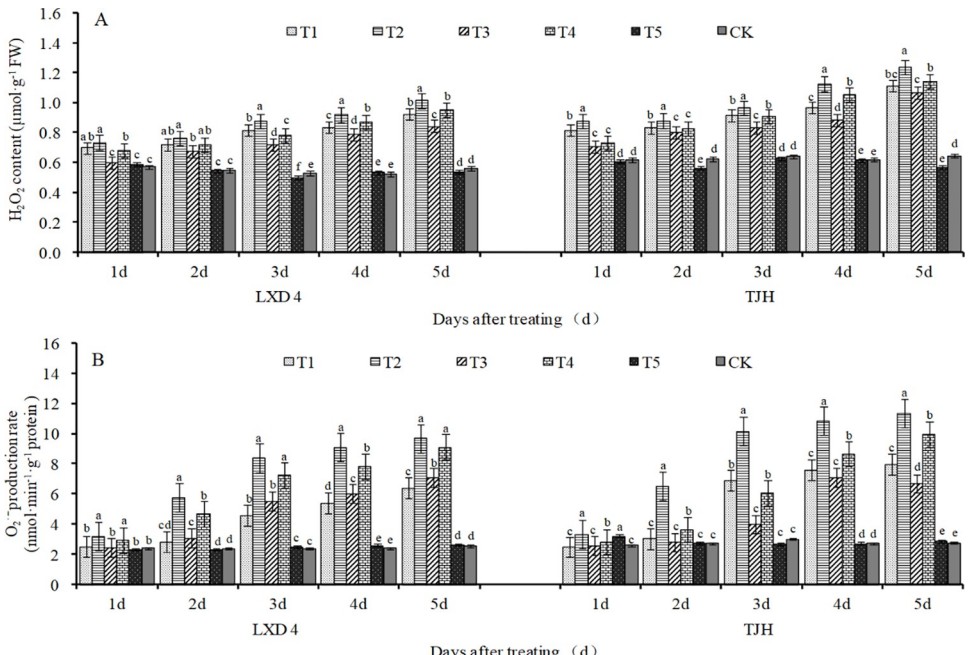

**Fig 3. Effect of exogenous ABA on O2·⁻—production rate and H2O2 content of adzuki bean leaves under chilling stress at flowering stage.** CK, plant grown in natural environment + spraying H2O; T1, plant grown in Average 12˚C + spraying exogenous ABA; T2, plant grown in Average 12˚C+ spraying H2O; T3, plant grown in Average 15˚C + spraying exogenous ABA; T4, plant grown in Average 15˚C+ spraying H2O; T5, plant grown in natural environment + spraying ABA. Note: Vertical bars represent standard error of the mean (n = 3). Different letters above the bars within each panel indicate significant differences across treatments according to the Duncan test (0.05).

tolerant variety LXD 4 has high enzyme activity in the cold-sensitive variety TJH. Compared with CK, low temperature caused the SOD, POD, and CAT activities of the two test varieties to significantly increase under any temperature treatment condition at each time. The effect of spraying exogenous ABA on cold-sensitive varieties is more obvious, especially the SOD and POD activities of TJH. At 1 to 5 days of treatment, TI is significantly higher than T2, and T3 is significantly higher than T4 (Fig 4A and 4B).

## Effect of exogenous ABA on SOD/CAT, SOD/POD in adzuki beans leaves under chilling stress

The ratio of SOD/CAT and SOD/POD can be used as a reference index for crops to resist stress under adversity conditions. The smaller the ratio, the stronger the ability to remove hydrogen peroxide and the weaker the damage to the crop. Low temperature induces lower SOD/CAT and SOD/POD of adzuki beans, especially the ratio of SOD/CAT as shown in Tables 2 and 3; when treated for 2–5 days, both varieties showed T2<T4<CK, indicating temperature under lower conditions, the stress response of adzuki beans is more intense. Foliar spraying with exogenous ABA can further reduce the ratio of SOD/CAT and SOD/POD, especially when treated for 3–5 days, both varieties showed T1<T2, T3<T4, respectively.

## Effect of exogenous ABA on AsA-GSH cycle non-enzymatic in adzuki beans leaves under chilling stress

As shown in Fig 5, the AsA, DHA, and AsA+DHA contents in the leaves of the two adzuki bean varieties showed a linear trend of increasing during the flowering stage with the extension

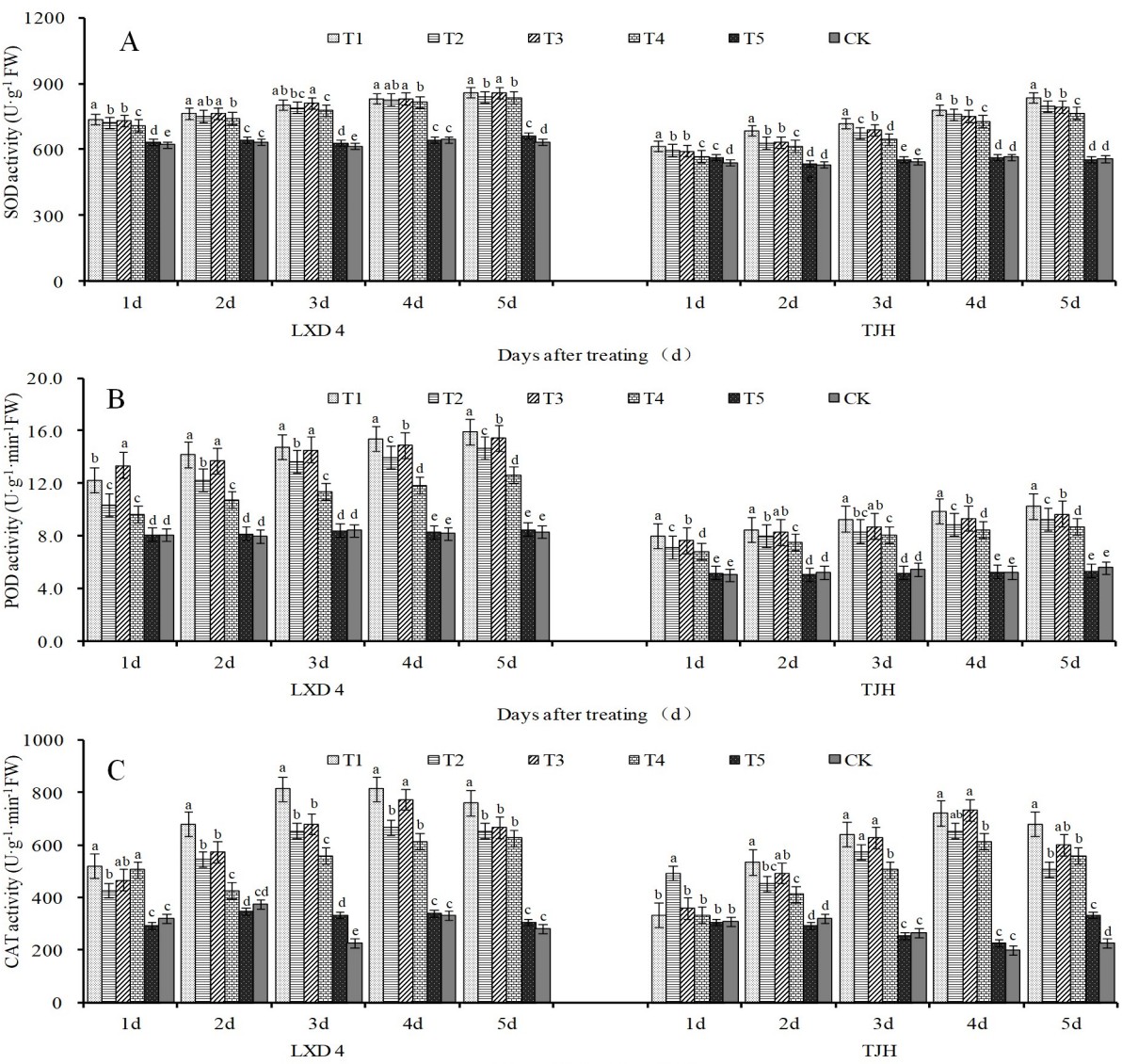

**Fig 4. Effect of exogenous ABA on enzymatic antioxidant system of adzuki beans leaves under chilling stress at flowering stage.** CK, plant grown in natural environment + spraying H2O; T1, plant grown in Average 12°C+ spraying exogenous ABA; T2, plant grown in Average 12°C+ spraying H2O; T3, plant grown in Average 15°C+ spraying exogenous ABA; T4, plant grown in Average 15°C+ spraying H2O; T5, plant grown in natural environment + spraying ABA. Note: Vertical bars represent standard error of the mean (n = 3). Different letters above the bars within each panel indicate significant differences across treatments according to the Duncan test (0.05).

of the low-temperature stress time. They were always higher than CK significantly. Under normal temperature conditions, foliar spraying with exogenous ABA can increase the content of AsA, DHA, and AsA+DHA, especially within 4 days after spraying, T5 is significantly higher than CK. Under low-temperature conditions, spraying with exogenous ABA can further increase ascorbic acid content. Among them, the control effect on the content of DHA, and AsA+DHA is more obvious. Both test varieties significantly showed T1>T2 and T3>T4 regardless of the average 12°C or the average 15°C when they were treated for 3–5 days (Fig 5B and 5C).

It can be seen from Fig 6 that, with the extension of the stress time, the contents of GSH, GSSG, and GSH+GSSG in T2 and T4 treatment leaves of both LXD 4 and TJH increased linearly (Fig 6A–6C), and T2 significantly higher than T4 at every processing time. At the same

**Table 2. Effect of exogenous ABA on SOD/POD of adzuki beans leaves under chilling stress at flowering stage.**

| Varieties | Treatment code | SOD/POD | | | | |
| --- | --- | --- | --- | --- | --- | --- |
| | | 1d | 2d | 3d | 4d | 5d |
| LXD 4 | T1 | 60.34±0.73 c | 53.93±0.32 d | 54.39±0.39 d | 54.24±0.54 d | 54.00±0.43 d |
| | T2 | 69.72±0.20 b | 61.47±1.53 c | 58.01±0.51 c | 59.19±0.51 c | 57.33±0.17 c |
| | T3 | 54.61±0.64 c | 55.74±1.03 d | 55.70±0.35 d | 55.76±0.37 d | 55.56±1.10 cd |
| | T4 | 74.26±3.88 ab | 69.29±0.57 b | 68.34±0.83 b | 68.65±0.31 b | 66.21±0.51 b |
| | T5 | 78.54±2.40 a | 78.76±1.22 a | 75.07±1.42 a | 77.83±0.75 a | 77.96±0.93 a |
| | CK | 77.29±0.68 a | 79.72±1.84 a | 73.31±0.78 a | 78.93±0.77 a | 76.39±1.67 a |
| TJH | T1 | 76.83±0.44 c | 77.98±0.64 bc | 77.76±1.54 b | 79.07±1.07 b | 81.71±2.39 c |
| | T2 | 83.46±1.14 c | 78.95±1.21 bc | 80.88±0.73 b | 85.94±0.86 b | 86.03±1.21 bc |
| | T3 | 77.74±1.26 c | 76.16±1.09 c | 79.09±1.37 b | 80.67±1.02 b | 82.18±1.00 c |
| | T4 | 83.37±0.89 b | 81.93±1.86 b | 80.02±1.62 b | 86.44±0.92 b | 88.52±1.19 b |
| | T5 | 109.35±2.98 a | 105.99±0.50 a | 107.31±3.44 a | 107.91±4.49 a | 104.54±3.16 a |
| | CK | 107.26±1.20 a | 102.04±3.86 a | 100.75±4.15 a | 109.19±4.35 a | 100.05±2.62 a |

Note: CK, plant grown in natural environment + spraying $H_2O$; T1, plant grown in Average 12°C+ spraying exogenous ABA; T2, plant grown in Average 12°C + spraying $H_2O$; T3, plant grown in Average 15°C+ spraying exogenous ABA; T4, plant grown in Average 15°C+ spraying $H_2O$; T5, plant grown in natural environment + spraying ABA. Different letters in the general column indicate a significant difference ($p<0.05$).

time, cold-tolerant varieties are higher than cold-sensitive varieties. It reached the maximum value at 5d after treating, of which the above-mentioned indicators of LXD 4 in T2 significantly increased by 40.24%, 83.37%, and 56.20%, respectively, compared with CK, and in T4 significantly increased by 35.95%, 82.20%, and 49.61%. The above-mentioned indicators in T2 and T4 treatment of TJH significantly increased by 58.26%, 116.21%, 79.23%, 46.28%, 111.90%, and 70.02% compared with CK, respectively. It can be seen from Fig 6B that exogenous spraying of ABA could further promote the increase of GSH, GSSH, and GSH+GSSG content in the leaves of both of the varieties, especially the content of GSSG has a significant increase under average 12°C. Compared with T2, the GSSG content in T1 both of the two varieties always increased, and the indicator of LXD 4 significantly increased by 13.86%, 15.25%,

**Table 3. Effect of exogenous ABA on SOD/CAT of adzuki beans leaves under chilling stress at flowering stage.**

| Varieties | Treatment code | SOD/CAT | | | | |
| --- | --- | --- | --- | --- | --- | --- |
| | | 1d | 2d | 3d | 4d | 5d |
| LXD 4 | T1 | 1.42±0.10 c | 1.13±0.06 c | 0.99±0.01 e | 1.03±0.04 d | 1.13±0.02 b |
| | T2 | 1.71±0.09 b | 1.38±0.03 b | 1.22±0.05 cd | 1.24±0.02 c | 1.29±0.05 b |
| | T3 | 1.57±0.06 bc | 1.34±0.05 b | 1.20±0.04 d | 1.08±0.04 d | 1.29±0.07 b |
| | T4 | 1.41±0.06 c | 1.75±0.08 a | 1.40±0.05 c | 1.33±0.02 c | 1.34±0.03 b |
| | T5 | 2.17±0.09 a | 1.86±0.04 a | 1.90±0.05 b | 2.68±0.03 a | 2.17±0.08 a |
| | CK | 1.97±0.12 a | 1.73±0.13 a | 0.99±0.15 a | 1.94±0.06 b | 2.29±0.17 a |
| TJH | T1 | 1.88±0.15 a | 1.29±0.07 c | 1.13±0.01 b | 1.09±0.08 b | 1.23±0.10 c |
| | T2 | 1.22±0.11 b | 1.39±0.06 c | 1.18±0.07 b | 1.17±0.07 b | 1.58±0.02 bc |
| | T3 | 1.67±0.07 a | 1.31±0.07 c | 1.10±0.05 b | 1.03±0.09 b | 1.33±0.04 c |
| | T4 | 1.73±0.17 a | 1.51±0.02 bc | 1.28±0.08 b | 1.20±0.13 b | 1.40±0.13 bc |
| | T5 | 1.92±0.23 a | 1.88±0.12 a | 2.26±0.26 a | 2.52±0.12 a | 1.67±0.03 b |
| | CK | 1.78±0.11 a | 1.67±0.10 ab | 2.06±0.12 a | 2.90±0.26 a | 2.52±0.22 a |

Note: Different letters in the general column indicate a significant difference ($p<0.05$).

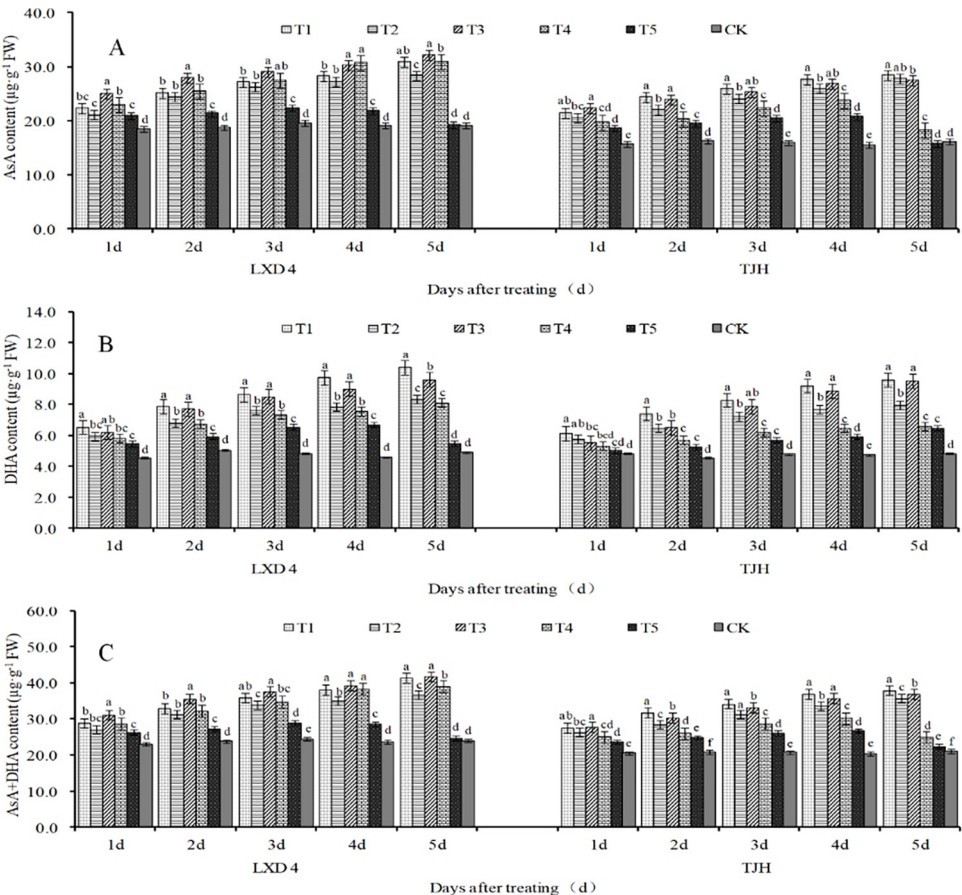

**Fig 5. Effect of exogenous ABA on ascorbic acid content of adzuki bean leaves under chilling stress at flowering stage.** CK, plant grown in natural environment + spraying H2O2; T1, plant grown in Average 12˚C+ spraying exogenous ABA; T2, plant grown in Average 12˚C+ spraying H2O2; T3, plant grown in Average 15˚C+ spraying exogenous ABA; T4, plant grown in Average 15˚C+ spraying H2O2; T5, plant grown in natural environment + spraying ABA. Note: Vertical bars represent standard error of the mean (n = 3). Different letters above the bars within each panel indicate significant differences across treatments according to the Duncan test (0.05).

26.60%, 20.49%, and 20.02% at 1–5 days after treatment, and TJH respectively increased by 10.00%, 18.21%, 18.26%, 59.26%, and 10.56%, significantly.

## Effect of exogenous ABA on AsA-GSH cycle enzymatic activities in adzuki beans leaves under chilling stress

Low temperature caused changes in APX activity in the leaves of the adzuki bean, showing a trend of initially increasing and then decreasing. Both varieties reached the highest value on the 4th day after the low-temperature treatment. Compared with the respective CK, the T2 and T4 of LXD 4 significantly increased by 91.75% and 68.23%, and the T2 and T4 treatments of TJH significantly increased by 85.69% and 31.28%, respectively. Under low-temperature conditions, spraying exogenous ABA can further increase the activity of APX, especially on the 2–4 days of treatment, for LXD 4, T1 was significantly increased by 19.56%, 12.78%, and 9.97% compared with T2; T3 was significantly increased by 22.78%, 20.75%, and T4 compared with T4, respectively. And for TJH, T1 is significantly increased by 22.78%, 20.75%, compared with T2, T3 is significantly increased by 30.39%, 19.53%, and 34.13%, compared with T4, respectively (Fig 7A).

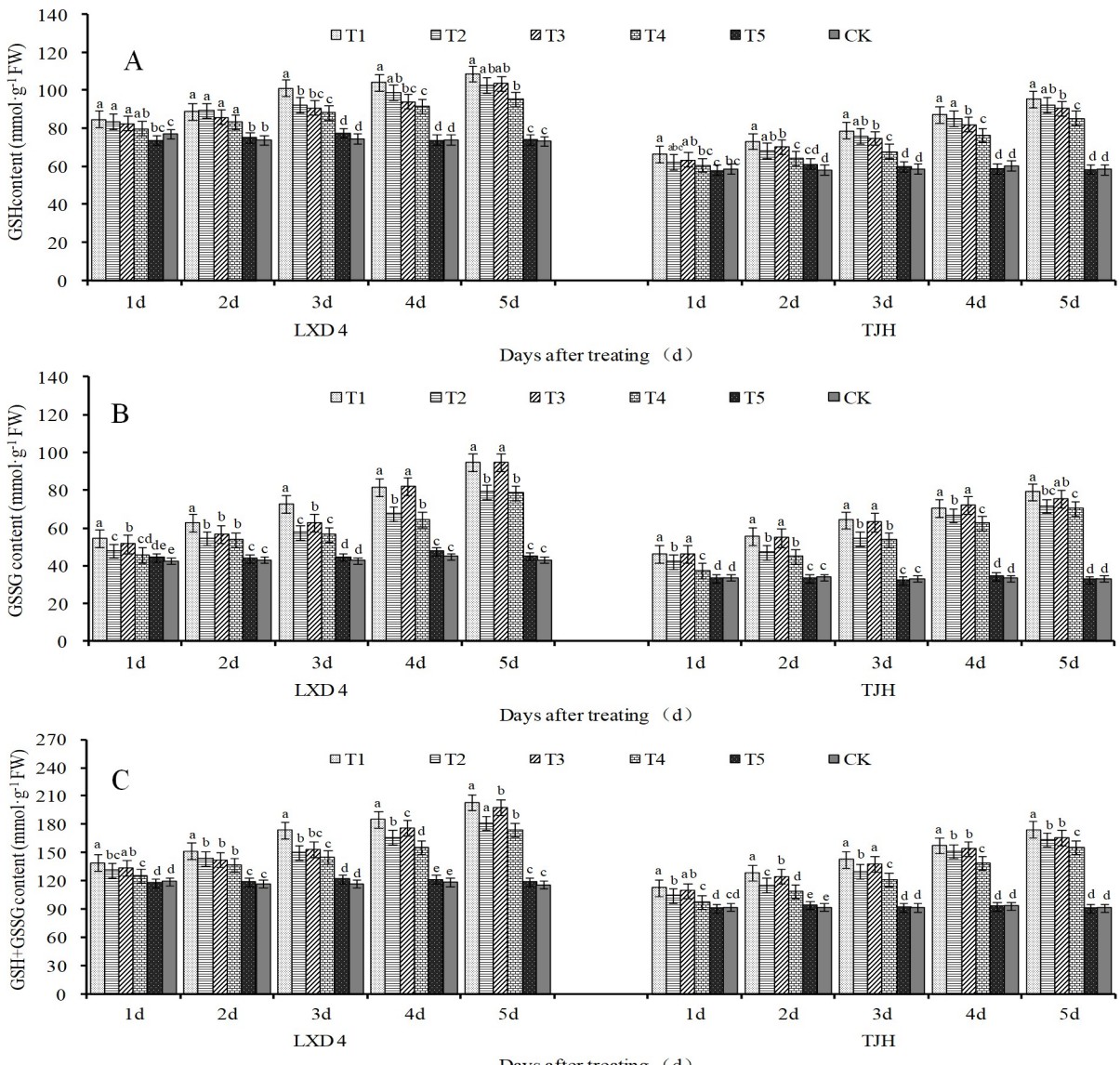

**Fig 6. Effect of exogenous ABA on adzuki bean leaves glutathione content under chilling stress at flowering stage.** CK, plant grown in natural environment + spraying H2O; T1, plant grown in Average 12°C+ spraying exogenous ABA; T2, plant grown in Average 12°C + spraying H2O; T3, plant grown in Average 15°C+ spraying exogenous ABA; T4, plant grown in Average 15°C+ spraying H2O; T5, plant grown in natural environment + spraying ABA. Note: Vertical bars represent standard error of the mean (n = 3). Different letters above the bars within each panel indicate significant differences across treatments according to the Duncan test (0.05).

The low temperature increased the GR activities in adzuki bean leaves, especially after 2 days of treatment. T2 was significantly higher than CK, and T4 was significantly higher than CK of both the test varieties (Fig 7B). Spraying exogenous ABA further improves GR activities. For LXD, under the average temperature of 15°C, T3 is significantly increased by 10.53%, 20.00%, 14.55%, 10.00%, and 6.92% compared to T4. For TJH, under the average temperature of 12°C, when treated for 2–5 days, compared with T2, T1 was significantly increased by 13.04%, 10.10%, 13.73%, and 11.32%, respectively (Fig 7B).

It can be seen from Fig 7C that low temperature caused a linear increase of DHAR activities in adzuki bean leaves. For both varieties, from the 2nd day to the 5th day of treatment, T2 and

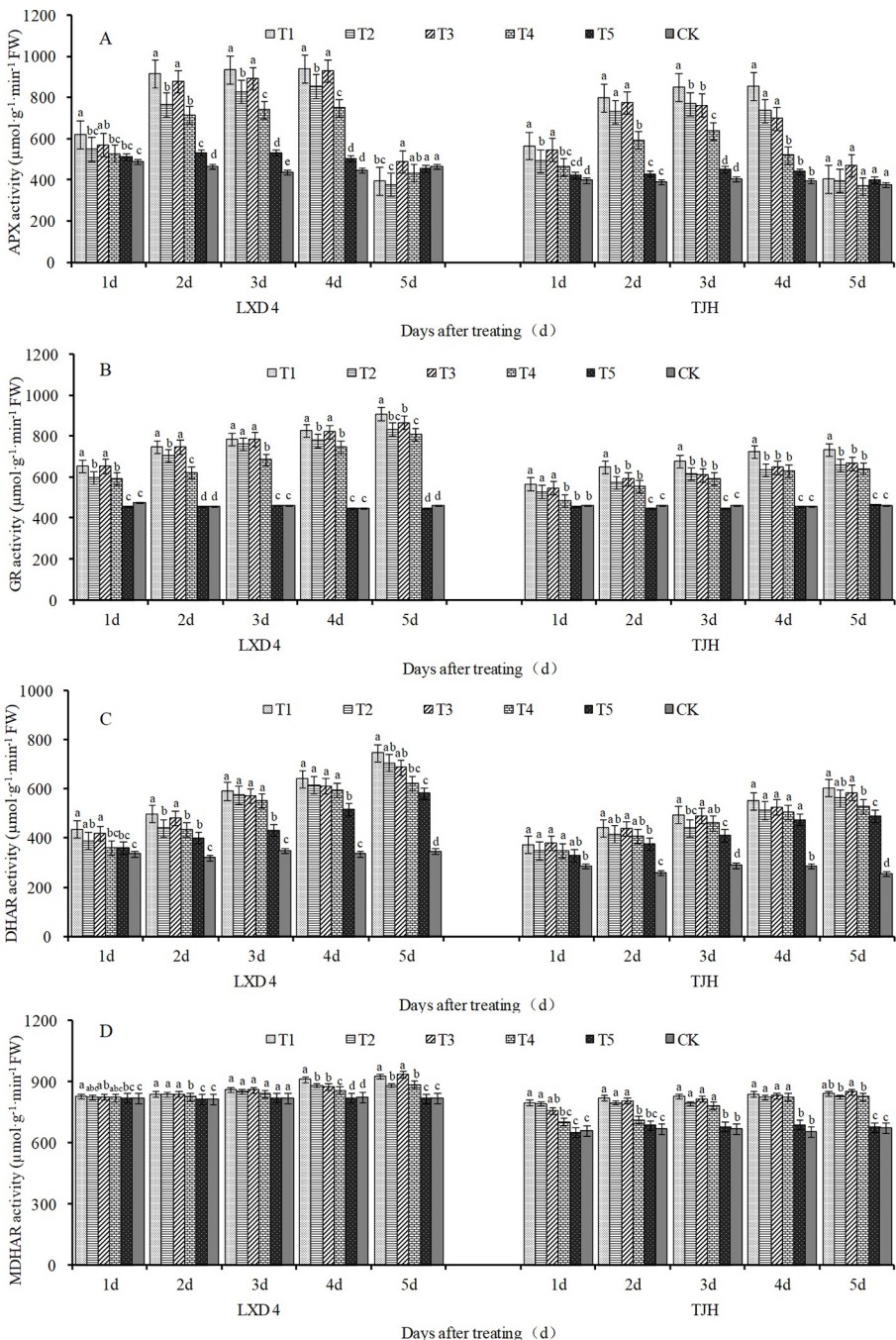

**Fig 7. Effect of exogenous ABA on key enzyme activities in the AsA-GSH defense system of adzuki bean leaves under chilling stress at flowering stage.** CK, plant grown in natural environment + spraying H2O; T1, plant grown in Average 12˚C+ spraying exogenous ABA; T2, plant grown in Average 12˚C+ spraying H2O; T3, plant grown in Average 15˚C+ spraying exogenous ABA; T4, plant grown in Average 15˚C+ spraying H2O; T5, plant grown in natural environment + spraying ABA. Note: Vertical bars represent standard error of the mean (n = 3). Different letters above the bars within each panel indicate significant differences across treatments according to the Duncan test (0.05).

T4 were always significantly higher than CK. Foliar spraying exogenous ABA can further increase the activity of DHAR. For LXD 4, T1 is 12.68%, 12.68%, 2.70%, 4.04%, and 5.73% higher than T2 from 1 to 5 days, and T3 is 16.38%, 10.71%, 3.95%, 3.14% and 10.50% higher

than T4, respectively. For TJH, treatment from 1 to 5 days, T1 is 7.14%, 5.97%, 11.97%, 7.27%, and 7.18% higher than T2, respectively, and T3 is 8.93%, 7.63%, 6.76%, 4.32%, and 10.59% higher than T4, respectively.

As shown in Fig 7D, low temperatures caused an increase in MDHAR activities in adzuki bean leaves. The cold-sensitive variety TJH is more obvious. At an average temperature of 12°C, T2 is significantly increased by 19.91%, 20.72%, 20.04%, 24.49%, and 25.13% compared with CK. At an average temperature of 15°C, T4 increased by 6.16%, 7.77%, 18.38%, 24.79%, and 25.09% compared with CK. Spraying exogenous ABA can further increase the activity of MDHAR. After treatment for 1 to 5 days, T1 was significantly increased by 20.63%, 24.11%, 25.26%, 27.13% and 27.51% compared with CK, and T3 was significantly increased by 14.65%, 22.03%, 23.35%, 25.90%, and 28.70% compared with CK.

## Effect of exogenous ABA on adzuki beans yield under chilling stress

Table 4 shows that low temperatures during the flowering stage can significantly affect the yield of adzuki beans. For LXD 4 or TJH, compared with CK, the yield shows CK>T4>T2 at each time, indicating that the lower the temperature, the greater the impact on the yield (Table 4). In addition, under the same low temperature, with the extension of the treatment time, the greater the degree of yield loss, T2 of LXD 4, compared with lower-temperature 1d, the yield at low-temperature 5 d decreased by 3.01g per plant, a decrease of 57.44%; while T4 treatment has dropped by 47.10%, reaching 2.52 g per plant. In T2 treatment for TJH, compared with lower-temperature 1d, the yield at low-temperature 5d decreased by 2.38g per plant, a decrease of 61.18%; while in T4 treatment, the output has dropped by 41.01%, reaching 1.71 g per plant.

Under normal temperatures, spraying exogenous ABA can increase adzuki bean yield. Compared with CK, T5 of LXD 4 was significantly increased by 0.77g per plant, an increase of 13.34%, and the T5 treatment of TJH was significantly increased by 1.30g per plant compared with CK, and the increase was 28.57%. Under low-temperature conditions, spraying exogenous ABA can inhibit adzuki bean production reduction. For LXD 4, at an average of 12°C, T1 was significantly higher than T2 by 12.06%, 15.42%, 14.94%, 22.62%, and 34.08% during

**Table 4. Effect of exogenous ABA on the yield of adzuki beans under chilling stress at flowering stage (unit: g·plant⁻¹).**

| Varieties | Treatment code | 1 d | 2 d | 3 d | 4 d | 5 d |
|---|---|---|---|---|---|---|
| LXD 4 | T1 | 5.87±0.10b | 5.27±0.07bc | 4.77±0.01cd | 4.12±0.10cd | 2.99±0.27c |
| | T2 | 5.24±0.09c | 4.54±0.12d | 4.15±0.05e | 3.36±0.09e | 2.23±0.04d |
| | T3 | 5.76±0.06bc | 5.33±0.07bc | 5.11±0.04c | 4.36±0.04c | 3.51±0.05c |
| | T4 | 5.35±0.06bc | 5.04±0.08cd | 4.50±0.05de | 3.76±0.08de | 2.83±0.10cd |
| | T5 | 6.54±0.35a | 6.79±0.33a | 6.71±0.43a | 6.50±0.40a | 6.58±0.30a |
| | CK | 5.77±0.12bc | 5.78±0.28b | 5.69±0.30b | 5.63±0.19b | 5.63±0.23b |
| TJH | T1 | 4.13±0.02cd | 3.93±0.03cd | 3.74±0.13cd | 3.20±0.20d | 2.51±0.13d |
| | T2 | 3.89±0.07d | 3.55±0.11d | 2.81±0.16e | 2.18±0.16e | 1.51±0.14e |
| | T3 | 4.53±0.09bc | 4.30±0.06bc | 3.98±0.04c | 3.84±0.10c | 3.22±0.14c |
| | T4 | 4.17±0.05bcd | 3.55±0.12d | 3.34±0.14d | 2.84±0.08d | 2.46±0.29d |
| | T5 | 5.85±0.22a | 5.94±0.17a | 5.90±0.49a | 5.77±0.04a | 5.79±0.19a |
| | CK | 4.55±0.12b | 4.76±0.52b | 4.82±0.52b | 4.77±0.19b | 4.53±0.08b |

Note: The data is the average value of ten repetitions; CK, plant grown in natural environment + spraying $H_2O$; T1, plant grown in Average 12°C+ spraying exogenous ABA; T2, plant grown in Average 12°C+ spraying $H_2O$; T3, plant grown in Average 15°C+ spraying exogenous ABA; T4, plant grown in Average 15°C+ spraying $H_2O$; T5, plant grown in natural environment + spraying ABA. Different letters in the general column indicate a significant difference ($p<0.05$).

treatment from 1 to 5 days. At an average of 15°C, T1 was higher than T2. by 7.66%, 5.75%, 13.56%, 15.96%, and 24.03%, respectively; the results of analysis of variance showed that the difference between the two reached a significant level ($p<0.05$) after treatment from 3 to 5 days. For TJH, under the condition of average 12°C, Compared with T2, T1 increased by 6.17%, 10.70%, 33.10%, 46.79%, and 66.23% during treatment from 1 to 5 days, the difference between the two reached a significant level ($p<0.05$) after treatment from 3 to 5 days. At an average of 15°C, T1 was higher than T2. by 8.63%, 21.13%, 19.16%, 35.21%, and 30.89%, respectively; the results of analysis of variance showed that the difference between the two reached a significant level ($p<0.05$) after treatment from 2 to 5 days.

## Discussion

Temperature is a crucial meteorological factor affecting crop growth and development [24], and low temperature during the growth period induces significant alterations in plant physiological metabolism [26]. Different varieties of the same crop have exhibit diverse responses to low temperature, while varying durations of low temperature exposure have distinct effects on the same variety [27]. This study found a positive correlation between the degree of yield reduction and chilling damage severity (Table 4 and Fig 8). Exogenous ABA has the function of regulating the anti-stress physiology of plants, and has multiple paths in resisting low temperature stress [28]. Its core mechanism involves diminishing reactive oxygen species accumulation, reducing membrane peroxidation, and minimizing cell membrane damage. The main stress response is to enhance the activity of antioxidant enzymes, increase the content of non-enzymatic antioxidant substances, etc., to adjust the reduction of MDA and $H_2O_2$ content and reduce the accumulation of $O_2^{-\cdot}$ and other ROS substances, and ultimately mitigate the impact of chilling damage on the yield of adzuki beans (Fig 9) [29–31].

### Effect of exogenous ABA on the balance of ROS

$H_2O_2$ is an important type of ROS in plants. Under normal conditions, the production and removal of $H_2O_2$ maintain a fine balance, which not only ensures that $H_2O_2$ in plants has certain physiological functions but also reduces its damage to plants. The effect is minimized. If this balance of $H_2O_2$ production and removal is broken, $H_2O_2$ accumulation occurs and eventually lead to a decline in crop yield (Fig 8). Because of its high redox activity, it can cause oxidative damage to macromolecules in cells, ultimately inhibit a variety of physiological and

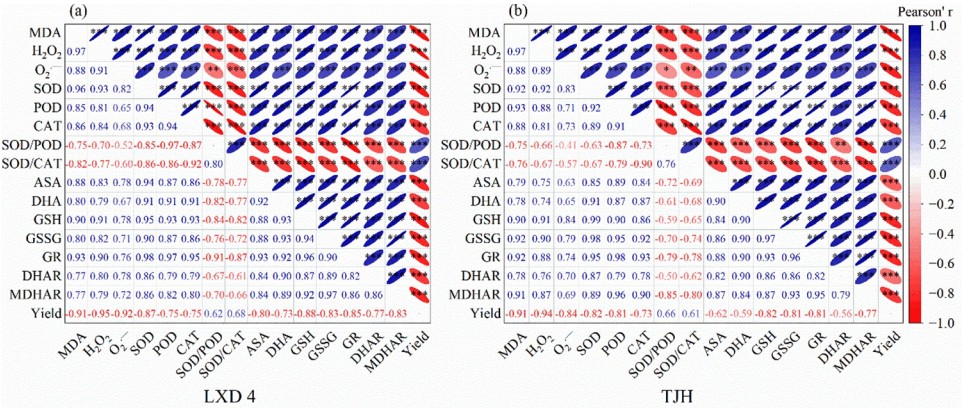

**Fig 8.** Pearson correlation analysis of key physiological indicators and yield in LXD 4(a) and TJH (b). *A significant difference at p < 0.05. ** A significant difference at p < 0.01. ***A significant difference at p < 0.001.

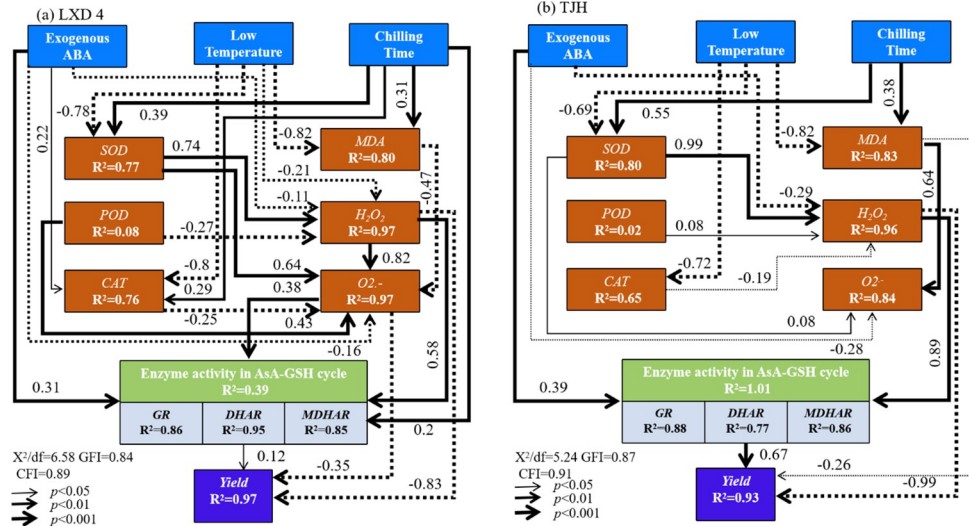

**Fig 9. Structural equation modeling (SEM) to examine the multivariate effects of exogenous ABA, low temperature and chilling time on yield of LXD 4 and TJH.** The solid and dashed lines indicate positive and negative coefficients, respectively; the thickness of the arrows indicates the magnitude of standardized path coefficient; R2 values indicate the proportion of variance explained for each endogenous variable. GFI, goodness of fit index; CFI, comparative fit index.

biochemical reactions in plants, and affect the regulation and metabolism of plants [32]. MDA serves as an indicator of cell membrane peroxidation, and its content correlates negatively with plant cold tolerance [33]. As stress increases, so does MDA accumulation. Low temperature can elevate the content of both $H_2O_2$ and MDA in plants while increasing plasma membrane permeability that damages cell membranes. Our study also found that low temperature exposure coupled with prolonged chilling time resulted in increased levels of $H_2O_2$ and MDA in LXD 4 and TJH. (Figs 2, 3 and 9).

Exogenous ABA has the physiological function of attenuating the increase of ROS and MDA content under adversity conditions. It can effectively mitigate the accumulation of MDA in crops under low-temperature stress, maintain the integrity of the membrane, and enhance the cold resistance of plants [11]. Our study shows that exogenous ABA can significantly reduce the accumulation of ROS substances of adzuki beans while increasing the ascorbic acid content to eliminate $H_2O_2$ and relieve low-temperature stress (Fig 9), which is consistent with the results of previous studies on the impact of exogenous ABA on soybean [34]. This may be attributed to the physiological function of exogenous ABA to improve the activity of protective enzymes, because when plants are in adversity, the balance system of free radical production and elimination in cells is destroyed, and there are many ways to eliminate free radicals in plants, the most important of which is the antioxidant enzyme system. Plants can maintain a low level of oxygen free radicals through the synergistic effects of SOD, POD, and CAT (Fig 9), which can slow down or defend against low-temperature stress to a certain extent [35].

There exists a significant positive correlation between SOD and superoxide anions ($O_2^{-\cdot}$) (Fig 8). The primary function of SOD is to remove superoxide anions ($O_2^{-\cdot}$), while simultaneously generating $H_2O_2$, which is subsequently degraded by POD and CAT through enzymatic action to prevent peroxidation damage caused by stress in plants (Fig 9). The increase in the ratio of SOD/POD and SOD/CAT will further increase the oxidative stress of plants [36]. This study discovered that chilling stress induced varying degrees of activity for protective

enzymes in adzuki beans (Fig 4A), which serves as the crop's defense mechanism against adverse conditions but only lasts for a short period. As the duration of stress increases, CAT activity begins to reduce (Fig 4C), this may be the critical threshold of its protection mechanism. Spraying exogenous ABA can further significantly increase the activity of protective enzymes, especially for SOD and CAT activities, and can effectively reduce the ratios of SOD/POD and SOD/CAT, indicating that spraying exogenous ABA can systematically coordinate the operation of the protective enzyme system, and can improve the antioxidant capacity of cells, alleviate or eliminate the peroxidation damage of ROS.

## Effect of exogenous ABA on the AsA-GSH cycle and yield

The AsA-GSH cycle is an important ROS scavenging system in plants and it plays an important role in effectively eliminating ROS [37]. AsA and GSH are essential non-enzymatic antioxidants in plants, serving as important participants in the AsA-GSH cycle. They can eliminate excessive ROS produced by plant cells due to stress damage and protect against the peroxidation of membrane lipids by free radicals [38]. AsA acts as an important antioxidant in plant photosynthetic tissues which can reduce the damage of oxidative stress to membranes and protect cell membrane permeability. The results of this experiment showed that the contents of AsA, DHA, and AsA+DHA in the leaves of the two adzuki beans varieties increased significantly with the extension of the low-temperature stress during the flowering period (Fig 5). Under normal temperature conditions, foliar spraying with exogenous ABA can increase the content of AsA, DHA, and AsA+DHA. In low-temperature conditions, spraying exogenous ABA can further increase the content of ascorbic acid. Notably, it exhibited effective control over DHA content (Fig 5B), especially for AsA+DHA (Fig 5C). Low-temperature stress caused a linear increase in the contents of GSH, GSSG, and GSH+GSSG in leaves,moreover, the lower the temperature result in higher levels of these three indicators(Fig 6A–6C). Exogenous spraying of ABA can additionally promote the increase of GSH, GSSH and GSH+GSSG content in the leaves of the two varieties, especially the GSSG content increased significantly at an average of 12˚C.

APX catalyzes the reaction between AsA and $H_2O_2$ to produce $H_2O$ and MDHA, which is the first step of the AsA-GSH cycle. The MDHA produced in this process can spontaneously form DHA [39] or be converted back to AsA by MDHAR with the assistance of APX in removing excess $H_2O_2$. Meanwhile, GPX also catalyzes GSH and $H_2O_2$ to generate $H_2O$ and GSSG. Under the catalysis of GR, GSSG can be reduced to GSH [40]. GR is also an important antioxidant enzyme that protects the sulfhydryl group of enzyme proteins. GR can regenerate AsA and further reduce oxidative stress [41]. DHAR acts as a bridge between AsA and DHA, using GSH as a substrate under NADH mediation to reduce DHA back to AsA while generating GSSG simultaneously. The results of our study showed that chilling stress promoted the increase of APX, GR, DHAR, and MDHAR activities in the leaves of the two adzuki beans varieties (Fig 7). Specifically, APX activity initially increased and then decreased over time while other three enzymes exhibited linear increases with lower temperatures leading to higher enzyme activities. Spraying exogenous ABA effectively promoted APX, GR, DHAR, and MDHAR activities to further increase, which can promote the AsA-GSH circulatory system to perform better physical and chemical functions, remove more ROS, and resist low-temperature damage to adzuki beans (Fig 9).

Exogenous application of ABA significantly increased crop yield factors under adverse stress conditions, and promoted an effective increase in crop productivity. [42] noted that exogenous ABA significantly increased the 100-seed weight and yield. [3] found that low temperatures during the seedling stage caused a significant reduction in the 100-seed weight and

the number of pods per plant, which in turn impacted the yield. However, spraying with exogenous ABA can significantly increase the 100-seed weight of adzuki beans after low-temperature stress and effectively alleviate the impact of low temperature on yield compared to clear water spray. Our study demonstrates that low temperature significantly decreases the grain weight of adzuki beans per plant. The yield loss gradually increases with the temperature decreases. Spraying exogenous ABA can significantly increase the grain weight per plant of adzuki beans after low-temperature stress by increasing the enzyme activity of the AsA-GSH cycle and effectively alleviating the impact of low temperature on yield (Fig 9).

## Conclusion

Chilling during the flowering stage causes changes in the physiological indicators of adzuki bean leaves. Low temperature causes a significant increase in the $H_2O_2$ and MDA content, as well as $O_2^{.-}$ production rate in adzuki bean leaves, promoting the increase of SOD and POD activities. The content of AsA and GSH substances was elevated, while crucial enzyme activities of the AsA-GSH cycle, such as GR, APX, etc., have been improved. The antioxidant enzyme activity and the change in the AsA-GSH cycle can cooperate to resist ROS-induced cell damage. Exogenous ABA has the function of resisting low temperatures and reducing crop damage by enhancing SOD, POD, and CAT activities in adzuki bean leaves; it also reduces the ratio of SOD/POD and SOD/CAT while increasing AsA and GSH substance contents to improve resistance against low-temperature stress for maintaining normal physiological activities. Low temperature during the flowering period caused a decrease in yield factors with a significant reduction in grain weight per plant of adzuki beans. However, under low temperature, compared with spraying with water, exogenous ABA can significantly increase the grain weight of adzuki beans per plant, effectively alleviating the impact of low temperature on yield. This experiment provides a theoretical basis for further research on the resistance of adzuki beans against low-temperature damage during the flowering stage. Furthermore, it highlights that exogenous ABA can alleviate damages caused by low temperatures, ensuring stable production of adzuki beans.

## Supporting information

**S1 Data.**
(ZIP)

## Author Contributions

**Data curation:** Shiya Wang, Qiulai Song, Yuqiang Liu, Qingjuan Wang.

**Formal analysis:** Xueyang Wang.

**Methodology:** Ning He, Xiaohui Dong, Shuqiang Chen, Wan Li.

**Software:** Deming Wang.

**Writing – original draft:** Hongtao Xiang.

**Writing – review & editing:** Xiaoyan Liang.

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
