## [Decision Letter · Decision Letter 0]

20 Oct 2023

PONE-D-23-26701Foliar Spraying Exogenous ABA: An Important Method to Resist Chilling Stress on Adzuki Beans (Vigna angularis) at the Flowering StagePLOS ONE

Dear Dr. Li,

Thank you for submitting your manuscript to PLOS ONE. After careful consideration, we feel that it has merit but does not fully meet PLOS ONE’s publication criteria as it currently stands. Therefore, we invite you to submit a revised version of the manuscript that addresses the points raised during the review process.

We look forward to receiving your revised manuscript.

Kind regards,

Umakanta Sarker

Academic Editor

PLOS ONE

Journal Requirements:

"This study was supported by Heilongjiang key R & D Program project (GA21B009-14), the earmarked fund for (CARS-08-G08)."

"This study was supported by Heilongjiang key R & D Program project (GA21B009-14), the earmarked fund for (CARS-08-G08)."

"This study was supported by Heilongjiang key R & D Program project (GA21B009-14), the earmarked fund for (CARS-08-G08)."

"The authors declare that they have no known competing financial interests or personal relationships that could have appeared to influence the work reported in this paper."

Reviewers' comments:

Reviewer's Responses to Questions

**Comments to the Author**

1. Is the manuscript technically sound, and do the data support the conclusions?

Reviewer #1: Yes

Reviewer #2: Yes

2. Has the statistical analysis been performed appropriately and rigorously? 

Reviewer #1: Yes

Reviewer #2: No

3. Have the authors made all data underlying the findings in their manuscript fully available?

Reviewer #1: Yes

Reviewer #2: Yes

4. Is the manuscript presented in an intelligible fashion and written in standard English?

Reviewer #1: Yes

Reviewer #2: Yes

5. Review Comments to the Author

Reviewer #1: This manuscript deals with an interesting topic and its results are interesting for the readers, however, there is a need to make corrections that are placed in the manuscript file as a comment. Please authors either make corrections or provide their answer.

Thanks

Reviewer #2: Statistical analysis results like significant levels, lettering or LSD values should mention in the tables. Some references are old. Updated and more reference will improve the manuscript. Discussion part need improvement.

6. PLOS authors have the option to publish the peer review history of their article (what does this mean?). If published, this will include your full peer review and any attached files.

Reviewer #1: **Yes: **Hedayatollah Karimzadeh Soureshjani

Reviewer #2: No

---

## [Author Response · Author response to Decision Letter 0]

11 Dec 2023

Response to reviewer comments

Point 1. Foliar spraying exogenous ABA resist chilling stress on Adzuki beans (Vigna angularis)

RESPONSE: We have made changes to the title. (lines 1-2)

Point 2. Write full name of AsA and GSH

RESPONSE: As per suggestion, we added the full name of AsA and GSH (lines 39-40)

Point 3. “Adzuki bean, one of the most important legume crops for human consumption, is cultivated in China, Korea, Japan, the Philippines, and other Southeast Asian countries” add references.

RESPONSE: Thanks, we have added the references. (lines 47)

Point 4. “Low temperature adversely affects plant growth and metabolism, causing a large accumulation of reactive oxygen species (ROS) free radicals within plants” add references.

RESPONSE: Thanks, we have added the references. (lines 56)

Point 5. “Among them, low temperature during flowering is the most harmful.” Why and add references.

RESPONSE: Thanks, we have rewritten the sentences and added references. (lines 62-69)

Point 6. Shows statistical differences among the values of different treatments. (lime 218)

RESPONSE: Thanks, we reworked the data and added errors. (lines 218-222)

Point 7. “Its core mechanism involves diminishing reactive oxygen species accumulation, reducing membrane peroxidation, and minimizing cell membrane damage. The main stress response is to enhance the activity of antioxidant enzymes, increase the content of non-enzymatic antioxidant substances, etc., to adjust the reduction of MDA and H2O2 content and reduce the accumulation of O2—·and other ROS substances, and ultimately mitigate the impact of chilling damage on the yield of adzuki beans” add supporting references.

RESPONSE: Thanks, we have added the references. (lines 339)

Point 8. “Spraying exogenous ABA can significantly increase the grain weight per plant of adzuki beans after low-temperature stress by increasing the enzyme activity of the AsA-GSH cycle and effectively alleviating the impact of low temperature on yield (Figure 9).” add supporting references.

RESPONSE: Thanks, this is partly a summary based on our experiments. (lines 339)

---

## [Decision Letter · Decision Letter 1]

24 Jan 2024

PONE-D-23-26701R1Foliar spraying exogenous ABA resists chilling stress on Adzuki beans (Vigna angularis)PLOS ONE

Dear Dr. Li,

Thank you for submitting your manuscript to PLOS ONE. After careful consideration, we feel that it has merit but does not fully meet PLOS ONE’s publication criteria as it currently stands. Therefore, we invite you to submit a revised version of the manuscript that addresses the points raised during the review process.

**ACADEMIC EDITOR:**

**Address the previous comments of both reviewers accurately. Also address typos, Units (following ISI standard) and spacing throughout the whole text, Tables and figure captions, etc.**

We look forward to receiving your revised manuscript.

Kind regards,

Umakanta Sarker

Academic Editor

PLOS ONE

Reviewers' comments:

Reviewer's Responses to Questions

**Comments to the Author**

1. If the authors have adequately addressed your comments raised in a previous round of review and you feel that this manuscript is now acceptable for publication, you may indicate that here to bypass the “Comments to the Author” section, enter your conflict of interest statement in the “Confidential to Editor” section, and submit your "Accept" recommendation.

Reviewer #1: (No Response)

2. Is the manuscript technically sound, and do the data support the conclusions?

Reviewer #1: Yes

3. Has the statistical analysis been performed appropriately and rigorously? 

Reviewer #1: Yes

4. Have the authors made all data underlying the findings in their manuscript fully available?

Reviewer #1: Yes

5. Is the manuscript presented in an intelligible fashion and written in standard English?

Reviewer #1: Yes

6. Review Comments to the Author

Reviewer #1: I am writing to express my concern regarding the manuscript entitled “Foliar spraying exogenous ABA resists chilling stress on Adzuki beans (Vigna angularis)”. It is evident that almost none of my comments were edited by the authors. Therefore, I have two guesses: firstly, the comments have not reached the authors, and secondly, they are not willing to make changes, and in the second case, they need to state their reasons.

7. PLOS authors have the option to publish the peer review history of their article (what does this mean?). If published, this will include your full peer review and any attached files.

Reviewer #1: No

---

## [Author Response · Author response to Decision Letter 1]

29 Feb 2024

Response to reviewer comments

Point 1. Foliar spraying exogenous ABA resist chilling stress on Adzuki beans (Vigna angularis)

RESPONSE: We have made changes to the title. (lines 1-2)

Point 2. Write full name of AsA and GSH

RESPONSE: As per suggestion, we added the full name of AsA and GSH (lines 39-40)

Point 3. “Adzuki bean, one of the most important legume crops for human consumption, is cultivated in China, Korea, Japan, the Philippines, and other Southeast Asian countries” add references.

RESPONSE: Thanks, we have added the references. (lines 47, 461-463)

Point 4. “Low temperature adversely affects plant growth and metabolism, causing a large accumulation of reactive oxygen species (ROS) free radicals within plants” add references.

RESPONSE: Thanks, we have added the references. (lines 56, 470-472)

Point 5. “Among them, low temperature during flowering is the most harmful.” Why and add references.

RESPONSE: Thanks, we have rewritten the sentences and added references. (lines 62-69, 473-485)

Point 6. Shows statistical differences among the values of different treatments. (lime 218)

RESPONSE: Thanks, we reworked the data and corrected errors. (lines 210-222)

Point 7. Shows statistical differences among the values of different treatments (Table 3). (lime 308)

RESPONSE: Thanks, we reworked the data and corrected errors, and modified the results analysis section. (lines 299-307, 308-Table 3)

Point 7. “Its core mechanism involves diminishing reactive oxygen species accumulation, reducing membrane peroxidation, and minimizing cell membrane damage. The main stress response is to enhance the activity of antioxidant enzymes, increase the content of non-enzymatic antioxidant substances, etc., to adjust the reduction of MDA and H2O2 content and reduce the accumulation of O2—·and other ROS substances, and ultimately mitigate the impact of chilling damage on the yield of adzuki beans” add supporting references.

RESPONSE: Thanks, we have added the references. (lines 339, 535-542)

Point 8. “Spraying exogenous ABA can significantly increase the grain weight per plant of adzuki beans after low-temperature stress by increasing the enzyme activity of the AsA-GSH cycle and effectively alleviating the impact of low temperature on yield (Figure 9).” add supporting references.

RESPONSE: Thanks, this is partly a summary based on our experiments. (lines 339)

---

## [Decision Letter · Decision Letter 2]

22 Mar 2024

PONE-D-23-26701R2Foliar spraying exogenous ABA resists chilling stress on Adzuki beans (Vigna angularis)PLOS ONE

Dear Dr. Li,

Thank you for submitting your manuscript to PLOS ONE. After careful consideration, we feel that it has merit but does not fully meet PLOS ONE’s publication criteria as it currently stands. Therefore, we invite you to submit a revised version of the manuscript that addresses the points raised during the review process.

We look forward to receiving your revised manuscript.

Kind regards,

Umakanta Sarker

Academic Editor

PLOS ONE

**Additional Editor Comments:**

Address all comments that was raised on original manuscript (first submitted manuscript) by both reviewers (see the attachments of reviewers and address them). Make a response letter following all comments of both reviewers on the original manuscript separately (see the attachments of reviewers and address them).

Reviewers' comments:

Reviewer's Responses to Questions

**Comments to the Author**

1. If the authors have adequately addressed your comments raised in a previous round of review and you feel that this manuscript is now acceptable for publication, you may indicate that here to bypass the “Comments to the Author” section, enter your conflict of interest statement in the “Confidential to Editor” section, and submit your "Accept" recommendation.

Reviewer #1: (No Response)

2. Is the manuscript technically sound, and do the data support the conclusions?

Reviewer #1: (No Response)

3. Has the statistical analysis been performed appropriately and rigorously? 

Reviewer #1: (No Response)

4. Have the authors made all data underlying the findings in their manuscript fully available?

Reviewer #1: (No Response)

5. Is the manuscript presented in an intelligible fashion and written in standard English?

Reviewer #1: (No Response)

6. Review Comments to the Author

Reviewer #1: Hello

First, I would like to apologize for the delay in evaluating this manuscript.

This is the second time that a revised manuscript has been sent to me while none of my requested revisions have been made or responded to. Therefore, I have no desire to re-evaluate this manuscript and with respect to the authors, I do not consider it suitable for publication in PLOSE ONE and reject it.

7. PLOS authors have the option to publish the peer review history of their article (what does this mean?). If published, this will include your full peer review and any attached files.

Reviewer #1: No

---

## [Author Response · Author response to Decision Letter 2]

6 Apr 2024

Thank you for your patience and for the insightful comments. I would like to express my sincere apologies for overlooking the comments in the PDF attachment during our initial revision process. It was an unintentional oversight, and I assure you it was not our intention to disregard the valuable feedback provided. Upon realizing this mistake, we promptly reviewed the comments and have addressed each point meticulously in our latest revisio.

Point 1. The results are described descriptively, maybe it is better to express some results quantitatively.

RESPONSE: Thank you very much for your comments. We have add quantitative description in the abstract. (lines 40-41)

Point 2. When the name of the compounds is mentioned in the text for the first time, write the full name and then use the abbreviation.

RESPONSE: Thanks very much for your comments. As per suggestion, we added the full name of AsA and GSH (line 40)

Point 3. lt is better that the keywords are not exactly the same as the title words.

RESPONSE: Thanks, we have made changes to the title. (lines 1-2)

Point 4. Please change this title to something more appropriate

RESPONSE: Thanks, We are very sorry for the poorly written. We have changed the title as “Leaf spray treatment”. (line 87)

Point 5. Please state with which software the correlation plot was drawn

RESPONSE: Thank you very much for your comments. We are sorry for neglecting this point, in the revised manuscript, we have made the changes as suggestion. (lines 149-150)

Point 6. lsn't it better to use 4 times or 3.8 times?

RESPONSE: Thanks, in the revised manuscript, we have made the changes as suggestion. (lines 158)

Point 7. The average comparison test and the statistical level of the study should be added below the figures and tables.

RESPONSE: Thank you for your comments. We are sorry that we did not write clearly, we have made the changes as suggestion. (lines 169-170, 196-197, 214-215, 229-230, 232, 264-265, 271-272, 313-314)

---

## [Decision Letter · Decision Letter 3]

15 May 2024

Foliar spraying exogenous ABA resists chilling stress on Adzuki beans (Vigna angularis)

PONE-D-23-26701R3

Dear Dr. Li,

We’re pleased to inform you that your manuscript has been judged scientifically suitable for publication and will be formally accepted for publication once it meets all outstanding technical requirements.

Kind regards,

Umakanta Sarker

Academic Editor

PLOS ONE

Additional Editor Comments (optional):

Reviewers' comments:

Reviewer's Responses to Questions

**Comments to the Author**

1. If the authors have adequately addressed your comments raised in a previous round of review and you feel that this manuscript is now acceptable for publication, you may indicate that here to bypass the “Comments to the Author” section, enter your conflict of interest statement in the “Confidential to Editor” section, and submit your "Accept" recommendation.

Reviewer #3: (No Response)

2. Is the manuscript technically sound, and do the data support the conclusions?

Reviewer #3: Partly

3. Has the statistical analysis been performed appropriately and rigorously? 

Reviewer #3: Yes

4. Have the authors made all data underlying the findings in their manuscript fully available?

Reviewer #3: Yes

5. Is the manuscript presented in an intelligible fashion and written in standard English?

Reviewer #3: Yes

6. Review Comments to the Author

Reviewer #3: The author did not make the needed review comments

so the manuscript needs major revision as in the attached file

7. PLOS authors have the option to publish the peer review history of their article (what does this mean?). If published, this will include your full peer review and any attached files.

Reviewer #3: No

---

## [Editor Report · Acceptance letter]

8 Jul 2024

PONE-D-23-26701R3 

PLOS ONE

Dear Dr. Li, 

I'm pleased to inform you that your manuscript has been deemed suitable for publication in PLOS ONE. Congratulations! Your manuscript is now being handed over to our production team.

Kind regards, 

on behalf of

Professor Umakanta Sarker 

Academic Editor

PLOS ONE